# Experimental determination of the force of malaria infection reveals a non-linear relationship to mosquito sporozoite loads

**Maya Aleshnick[1], Vitaly V. Ganusov[2], Gibran Nasir[1], Gayane Yenokyan[3], Photini Sinnis[1] ***

1 Johns Hopkins Malaria Research Institute and Department of Molecular Microbiology & Immunology, Baltimore, Maryland, United States of America, 2 Departments of Microbiology and Mathematics, University of Tennessee, Knoxville Tennessee, United States of America, 3 Johns Hopkins Biostatistics Center, Department of Biostatistics, Johns Hopkins Bloomberg School of Public Health, Baltimore, Maryland, United States of America

* psinnis1@jhu.edu

**Data Availability Statement:** We are submitting a supplementary excel file that contains all of the data used to perform the analyses in this manuscript.

## Abstract

*Plasmodium* sporozoites are the infective stage of the malaria parasite. Though this is a bottleneck for the parasite, the quantitative dynamics of transmission, from mosquito inoculation of sporozoites to patent blood-stage infection in the mammalian host, are poorly understood. Here we utilize a rodent model to determine the probability of malaria infection after infectious mosquito bite, and consider the impact of mosquito parasite load, blood-meal acquisition, probe-time, and probe location, on infection probability. We found that infection likelihood correlates with mosquito sporozoite load and, to a lesser degree, the duration of probing, and is not dependent upon the mosquito's ability to find blood. The relationship between sporozoite load and infection probability is non-linear and can be described by a set of models that include a threshold, with mosquitoes harboring over 10,000 salivary gland sporozoites being significantly more likely to initiate a malaria infection. Overall, our data suggest that the small subset of highly infected mosquitoes may contribute disproportionally to malaria transmission in the field and that quantifying mosquito sporozoite loads could aid in predicting the force of infection in different transmission settings.

## Author summary

Malaria is a leading cause of death in many parts of the world. Infection is initiated when infected *Anopheles* mosquitoes inject sporozoites as they look for blood. Though transmission is a bottleneck for the parasite and thus a good point for intervention, many aspects of transmission remain poorly understood. In this study, using a rodent model of malaria, we found that the majority of infective bites do not result in malaria infection. Furthermore, we found that the bites of mosquitoes with heavy parasite burdens are significantly more likely to result in blood stage infection. These data have important implications for designing interventions targeting transmission stages of the malaria parasite as they

**Funding:** This work was funded by the National Institutes of Health R01 A1056840 and AI132359 (PS) and R01 GM118553 (VVG), and The Bloomberg Family Foundation (PS, MA, GN). We would also like to acknowledge support for the statistical analysis from the National Center for Research Resources and the National Center for Advancing Translational Sciences (NCATS) of the National Institutes of Health through Grant Number 1UL1TR001079. The funders had no role in study design, data collection and interpretation.

**Competing interests:** The authors have no competing interests.

suggest that reducing parasite loads, even without completely eliminating them, could be effective against disease spread. We also found that mosquitoes that probe but do not succeed in finding blood are equally likely to initiate infection, an important finding for human vaccine trials. Overall this work contributes to our understanding of the epidemiology of malaria and should aid in the development of malaria elimination strategies.

## Introduction

Malaria remains one of the most important infectious diseases in the world, responsible for approximately 200 million cases and 500,000 deaths annually [1], with the majority of deaths occurring in young children in sub-Saharan Africa. Protists of the genus *Plasmodium* are the causative agents of the disease and are transmitted by Anopheline mosquitoes. Sporozoites, the infective stage of the parasite, reside in mosquito salivary glands and are injected into the host's skin as the mosquito searches for blood [2–4]. From there sporozoites enter the blood circulation and travel to the liver where they invade hepatocytes and divide into thousands of hepatic merozoites (reviewed in [5, 6]). These liver stage parasites initiate the blood stage of infection, where iterative rounds of replication lead to high parasite numbers and clinical symptoms. The pre-erythrocytic stage of infection is not associated with clinical symptoms and is characterized by low parasite numbers whose goal it is to gain a foothold in the mammalian host. That this process is not always successful has been known for some time [7], however, it has been difficult to establish a more quantitative foundation of this transmission bottleneck [8].

Mathematical models of malaria transmission dynamics use the entomological inoculation rate (EIR), the number of infected bites per person per unit time, as the measure of the intensity of malaria transmission and do not consider the proportion of infected mosquito bites that result in patent blood-stage infection [9]. This omission was noted early on by malariologists who were struck by the discrepancy between the numbers of infective mosquito bites and human infections, and they added the parameter *b*, the proportion of infected mosquito bites that result in malaria infection, into transmission equations [8] (see Box 1). In contrast to the EIR, which has been extensively measured and reported, the proportion of infected mosquito bites that result in malaria infection has not been directly studied. Nonetheless, attempts to indirectly estimate "b" by back-calculating from measures of infant infection rates and EIR have arrived at estimates between 1 and 10% of infective bites leading to infection [7, 10, 11].

---

### Box 1

EIR = *maz*

*m* = the anopheline density in relation to humans

*a* = the average number of persons that one mosquito bites in one day

*z* = the proportion of mosquitoes that have sporozoites in their salivary glands

Force of infection = *mazb*

*b* = the proportion of infective mosquito bites that result in a malaria infection

---

In this study we use the rodent malaria model *Plasmodium yoelii* to measure the risk of infection after a single infected mosquito bite and correlate this to salivary gland sporozoite load, probe time, location of the bite, and whether the mosquito took a blood meal. We find that the likelihood of infection increases significantly when a mosquito harbors over 10,000 salivary gland sporozoites and correlates to a lesser degree, with the duration of mosquito probing. Notably, infection probability was not impacted by whether a blood meal had been acquired by the mosquito.

## Results

The rodent malaria parasite *P. yoelii* was used for these experiments because it provides a sensitive system to detect potentially infectious bites. Indeed, *P. yoelii* sporozoite infectivity for laboratory mice is similar to the infectivity of human and primate malaria sporozoites for their natural hosts [12–16]. Two to four *P. yoelii* sporozoites inoculated intravenously (IV) initiate malaria infection in 50% of mice [12–14], while 10 *P. vivax* or *P. cynomolgi* sporozoites initiated infection in 100% of human volunteers or rhesus monkeys, respectively [15, 16]. This is in contrast to another rodent malaria model, *Plasmodium berghei* where $ID_{50}$ values are 50 to 1,000 times higher [17, 18]. Experiments in which sporozoites are inoculated by *P. yoelii* and *P. falciparum* infected mosquitoes found that in both cases, 5 infected mosquito bites are required to infect 100% of mice or human subjects, respectively [14, 19, 20]. Here we used the *P. yoelii* rodent malaria model and performed 412 single mosquito feeds, noting the duration of mosquito probing and whether a blood meal was taken. A "mosquito feed" or "bite" was defined as an encounter in which the mosquito proboscis penetrated the skin and probing was initiated, as determined by direct observation of the proboscis. Subsequently we quantified the salivary gland load of each mosquito and followed each mouse for blood stage malaria infection. For consistency, feeds were performed on the ear pinna of the mouse, though in a separate group of experiments we tested whether the location of mosquito feeding impacted the outcome. The raw data from all feeds can be found in the Supplemental Table.

### The impact of blood meal acquisition on sporozoite transmission efficiency

Blood feeding by mosquitoes has two spatially and temporally distinct stages: searching for blood during which time mosquitoes are probing while salivating into the skin, and imbibing blood when salivation stops and mosquitoes take up blood via their food canal [21]. Thus, it is reasonable to hypothesize that a pathogen residing in mosquito salivary glands would be inoculated during probing, making its portal of entry the dermis rather than the blood circulation. Indeed, several lines of evidence demonstrate that this is the case for *Plasmodium* sporozoites [2–4, 22–24]. Nonetheless, some investigators continue to labor under the assumption that sporozoites are inoculated directly into the blood circulation. Because our goal was to determine the probability that a single infected mosquito bite would result in a malaria infection, we first determined whether blood feeding is an appropriate readout for exposure to sporozoites.

To test this, we compared the likelihood of malaria infection after mosquito feeds that resulted in the acquisition of blood versus those that did not. Mosquitoes were monitored by visual observation of blood in the midgut and follow-up microscopy examination of the esophagus for traces of blood, and mice were followed for blood stage malaria infection. As shown in Fig 1, 58% of mosquitoes succeeded in finding blood, a number that is consistent with previous findings [25]. Importantly, the likelihood of initiating an infection in the mouse was not dependent upon whether the mosquito found blood, with 16% and 14% of mice becoming

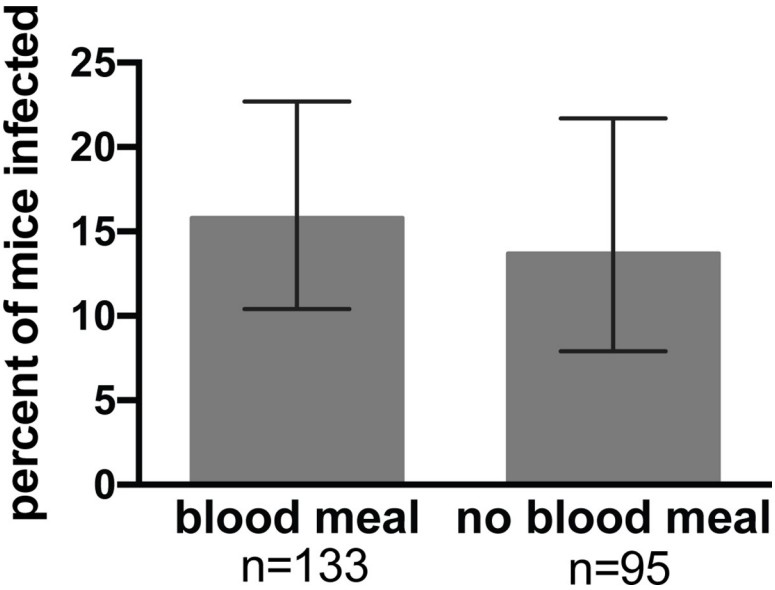

**Fig 1. Acquisition of a blood meal by the biting mosquito is not associated with the successful transmission of** ***Plasmodium.*** Single mosquitoes were allowed to probe on individual mice until they obtained a blood meal or lost interest in probing. Following this, mosquito midguts were inspected for the presence of blood, salivary gland sporozoite loads were measured, and mice were followed for blood stage malaria infection. Shown is the percent of mice that became infected after being probed upon by mosquitoes that were or were not successful in obtaining a blood meal. Error bars show 95% confidence intervals calculated using Jefferey's intervals for binomial distribution [66]. Logistic regression analysis of the probability of infection demonstrates that the acquisition of a blood meal has no impact on the likelihood of malaria infection (p = 0.705). n = total number of mice in each group. Shown are pooled data from 10 independent experiments with 10 to 45 mice per experiment. All experiments in which probe time was not controlled are included in this analysis for a total of 228 mosquito/mouse pairs.

positive for malaria infection after being probed upon by mosquitoes that found blood versus those that did not, respectively. Logistic regression analysis of malaria infection indicates that the risk of malaria is not associated with the blood meal status of the infecting mosquito (p = 0.705), and the estimated odds ratio of malaria infection after being bitten by a mosquito that took a blood meal versus one that did not is 1.18 (95% CI 0.5 to 2.8). These data expand upon the previous demonstration that probing is sufficient for transmission to occur [2, 4] and show that blood meal acquisition is not only noncompulsory, but also does not increase the likelihood of initiating an infection.

## The probability that a single infected mosquito bite will result in a malaria infection and its relationship to mosquito infection intensity

To determine the likelihood that a single infected mosquito bite would result in a malaria infection, we performed 412 single mosquito feeds and followed the mice for blood stage infection. Following each feed, mosquito salivary glands were removed and sporozoite load was determined by qPCR, using a sporozoite standard curve that spans five orders of magnitude (S1 Fig). Since previous studies have shown that mosquitoes inoculate only a small percentage of the sporozoites in their salivary glands, these measurements are a reasonable approximation of sporozoite loads when probing was initiated [22, 26–30]. From a total of 412 mice, each bitten by a single infected mosquito, 17.5% became positive for blood stage infection. These data demonstrate that the majority of infectious mosquito bites do not result in malaria infection.

One of the goals of our study was to assess the impact of mosquito salivary gland sporozoite load on the probability of blood stage infection. Since *Plasmodium* infections of laboratory

mosquito colonies are biased towards highly infected mosquitoes, in order to generate parasites for experiments we adjusted our protocol to infect mosquitoes with low and high numbers of *P. yoelii* gametocytes in order to have a range of sporozoite loads. The distribution of salivary gland loads for all mosquitoes in our dataset is shown in S1 Fig. The median load was 8,865 sporozoites, with a range of 1–647,714.

To determine whether the intensity of mosquito infection plays a role in the likelihood of initiating a malaria infection, we performed several complementary analyses on our data set. First, we binned salivary gland loads of the infecting mosquitoes on a log scale similar to that used to grade mosquito parasite burdens in field settings [27], and calculated the percent of mice infected for each group (Fig 2A). Plotted in this manner, the data suggest that there is a jump in infection likelihood when mosquitoes have salivary gland loads of ≥10,000 sporozoites. We further evaluated the relationship between salivary gland sporozoite load and the likelihood of infection by logistic regression analysis. Visual inspection of this relationship suggested that it is non-linear and we found that introducing a linear spline with a knot anywhere between 10,000 to 20,000 salivary gland sporozoites fit the data significantly better than a standard logistic model (likelihood ratio test, $\chi_2 = 27.04$, $p<0.001$). From these data we derived a predicted probability of malaria infection as a function of mosquito salivary gland sporozoite load, placing the knot at 20,000 sporozoites (Fig 2B). The relationship between salivary gland sporozoite load and the probability of malaria infection is non-linear: For sporozoite loads <20,000, the estimated odds of malaria goes up by 12% for every increase of 1,000 sporozoites (CI 5% to 19%, p<0.001) and for sporozoite loads ≥20,000, the estimated odds of malaria go up by 0.3% for every increase of 1,000 sporozoites or 3% increase for every increase of 10,000 sporozoites (CI 0.04% to 0.6%, p = 0.024). Thus, while salivary gland sporozoite load strongly correlates with infection probability, this effect levels off as salivary gland loads increase beyond 20,000 sporozoites.

Both the shape of the curve outlined by the binned data (Fig 2A) and the need for a linear spline in the logistic regression analysis (Fig 2B), suggested that these data may be best described by a step-function rather than a continuous line. We therefore evaluated the fit of several alternative mathematical models (see Materials and Methods for more detail), comparing a threshold model, in which infection probability is described by a step-like function of salivary gland load, to two commonly used continuous models, the single-hit and power-law models, in which infection probability increases as a linear or power function of salivary gland load, respectively. Using Akaike weights to directly compare these models, we found that a threshold model fits the data better than the two continuous models (Fig 2C).

The good fit of the data to both logistic-regression with a spline (Fig 2B) and the threshold model (Fig 2C) suggests that there is a rapid change in the slope of the infection probability curve between mosquito salivary gland loads of 10,000 and 20,000 sporozoites. However, while the threshold model incorporates a distinct "step" (Fig 2C), the logistic regression model shows a gradual increase in infection probability between salivary gland loads of ~ 10,000 to 20,000 sporozoites (S2 Fig). To determine whether a 'hard' or a more gradual 'soft' threshold better describes the data, we directly compared three alternative models that incorporate a rapid change in slope: the threshold model (Eq 3), a slope-threshold model which incorporates a linear slope prior to the threshold (Eq 4), and the double-logistic model (Eq 5; similar to the model shown in 2B). Based on Akaike weights, we found that the double-logistic and slope-threshold models fit the data somewhat better than a strict threshold model (S3 Fig). However, the differences in Akaike weights were small and all models fit the data reasonably well based on the Hosmer-Lemeshow goodness of fit test (p>0.1). Thus, though a softer threshold fits the data marginally better, all models that incorporate a rapid change in infection probability between 10,000 to 20,000 salivary gland sporozoites fit the data relatively well.

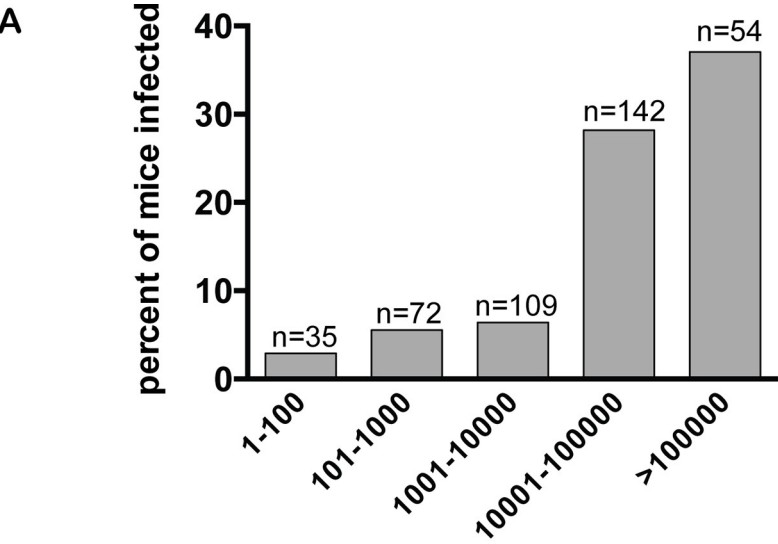

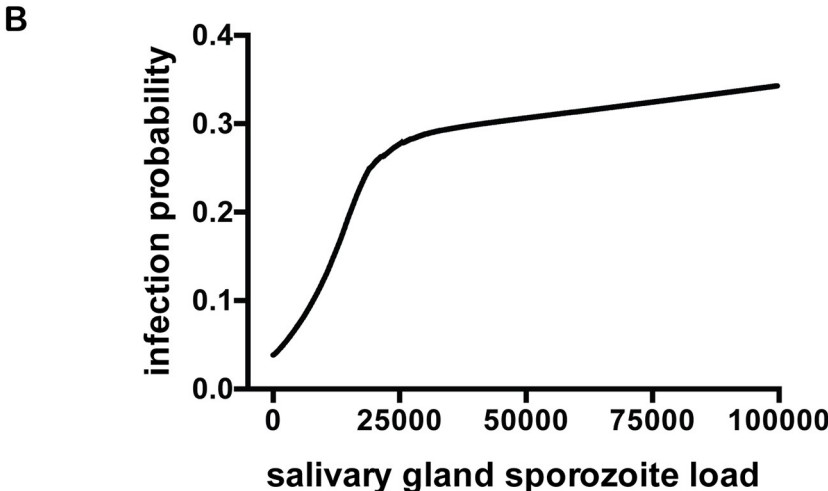

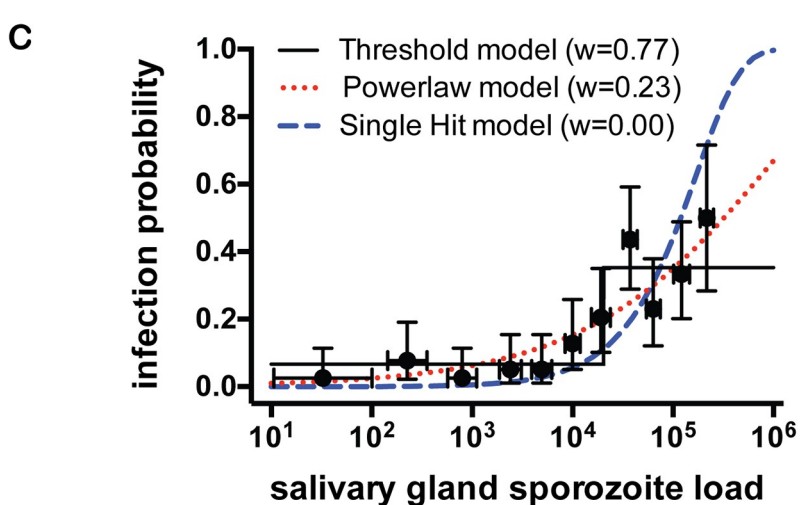

**Fig 2. Mosquito salivary gland sporozoite load significantly impacts the likelihood of malaria infection.** Single mosquito feeds were performed and subsequently salivary gland sporozoite loads were measured and mice were followed for blood stage malaria infection. Data are pooled from 20 independent experiments with 7 to 44 mice per experiment. Total n = 412 **(A) Log-binning of mosquito salivary gland loads and percent of infected mice in each bin.** Mosquitoes are binned using the traditional method of grading salivary gland infections on a log-scale and for each bin the percent of mice positive for malaria infection is plotted. Above each bar is the number of mouse-mosquito pairs in each bin (n). **(B) Logistic regression analysis of the probability of malaria infection as it relates to salivary gland sporozoite load.** Predicted probability of malaria infection, from logistic regression with a linear spline at 20,000 salivary gland sporozoites, is plotted against mosquito sporozoite load and shown is the smoothed Lowess curve. A statistically significant relationship between salivary gland load and probability of malaria was observed (p<0.001). The x-axis is cut at 100,000 to focus on sporozoite loads more commonly observed in the field. Logistic regression plotted for the full range of sporozoite loads in our dataset is shown in S2 Fig. **(C) Mathematical modeling of infection probability as it relates to salivary gland sporozoite load.** Three alternative models, (single hit, powerlaw, and threshold), were fit to the entire raw dataset (n = 412) and compared using Akaike Information Criterion to generate Akaike weights (w). The threshold model best describes the probability of malaria infection as a function of salivary gland sporozoite number (highest weight). The data points on the graph are shown for illustrative purposes: Raw data were binned with equal number of mosquitoes in each group (n = 41), except the last group, which has 43 mosquitoes. Error bars on the y-axis show 95% confidence intervals calculated using Jefferey's intervals for binomial distribution [66] and on the x-axis show 67% confidence intervals calculated using normal distribution. The single hit, powerlaw, and threshold models are described in the Materials and Methods by Eqs 1, 2 and 3, respectively) and fit was determined using the maximum likelihood method (Eq 6 in Materials and Methods).

Our data together with the common practice of grading sporozoite salivary gland loads in malaria-endemic areas on a log scale suggests that a cutoff of 10,000 salivary gland sporozoites might be a useful indicator for estimating the force of infection in different transmission settings. Indeed, binning of the mosquitoes in our study into those with salivary gland loads <10,000 or ≥10,000 sporozoites shows that infection likelihood is 5.6% versus 30.6%, respectively (S4 Fig). Logistic regression of malaria risk with salivary gland loads dichotomized at 10,000 sporozoites found that mosquitoes with gland loads >10,000 have a 7.5-fold higher odds of initiating infection (95% CI 3.6 to 15.8, p<0.001; S4 Fig). Using the infection probabilities from these dichotomized data we estimated "b" for different transmission settings (S5 Fig). Though the applicability of our study to the field remains to be tested, these results raise the possibility that mosquito salivary gland loads could be used in the field to estimate the force of infection.

## The duration of mosquito probing and its impact on infection probability

Sporozoites are transmitted to the mammalian host during mosquito probing [2], and while it has been suggested that an increase in the duration of probing increases the transmission potential of the parasite [31], there is some controversy on this matter [29–32]. To investigate this, we performed single mosquito feed experiments in which probe time was controlled. Mosquitoes were allowed to probe for 10 seconds, 1 minute or 5 minutes, after which the mice were followed for malaria infection. Mosquitoes that were allowed to probe longer were more likely to initiate a malaria infection (Fig 3; p = 0.020, Fisher's exact test) with pairwise comparisons showing a statistically significant difference in infection probability between the 10 sec and 5 min probe times (p = 0.021) and between the 1 min and 5 min probe times (p = 0.025). Because salivary gland load impacts the likelihood of malaria infection, we wanted to ensure that the intensity of infection was not a confounding factor in these experiments. The number of mosquitoes with ≥10,000 salivary gland sporozoites in each probe time group was approximately equivalent, with the 10 sec group having slightly more mosquitoes with ≥10,000 sporozoites (72%) than the 1 min (58%) and 5 min (60%) groups. After adjusting for sporozoite load in a multivariable logistic regression model, the likelihood of infection was still higher in the 5 min compared to the 10 sec group (estimated odds ratio of malaria infection = 6.7, 95% CI 2.1

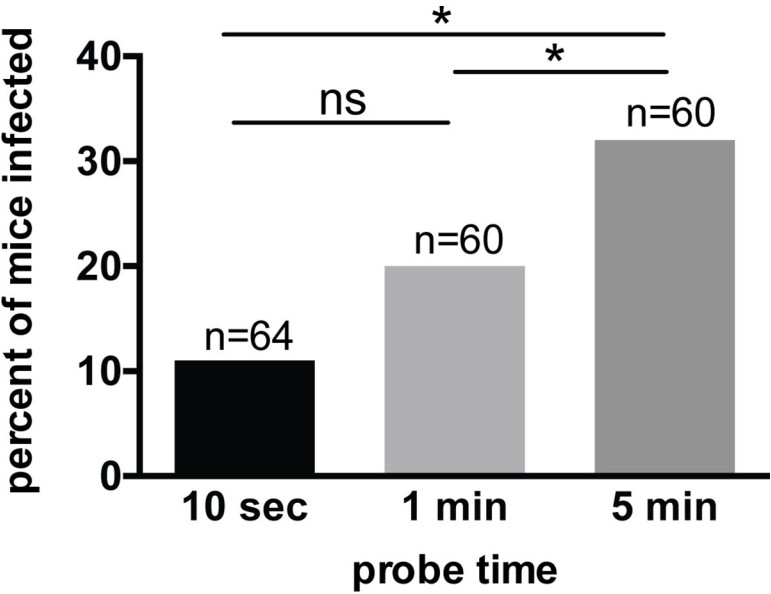

**Fig 3. Mosquito probe time impacts the likelihood of malaria infection.** Single-mosquito feeds were performed with the duration of probing experimentally manipulated at 10 seconds, 1 minute, or 5 minutes. Following this, salivary gland sporozoite loads were measured and mice were followed for blood stage malaria infection. There was a statistically significant association between probe time and probability of malaria infection (Fisher's exact test p-0.020). Pair-wise comparisons were performed using logistic regression with odds of malaria infection as the outcome, probe time as a categorical predictor and robust variance. The results indicated statistically significant differences between 10 sec vs 5 min (p = 0.021) and 1 min vs 5 min (p = 0.025); while the comparison between the 10 sec vs 1 min groups did not reach statistical significance (p = 0.275). n = total number of mice in each group, pooled from 6 independent experiments with 10 to 15 mice per group per experiment. ns = not significant. See Methods and Materials for details on experimental design and statistical analyses.

to 21.8, p = 0.002), further supporting an association between mosquito probe time and infection probability.

## The location of mosquito probing and its impact on the likelihood of malaria infection

After their inoculation, sporozoites move in the dermis to locate a blood vessel for entry into the bloodstream and subsequent transport to the liver [23, 24]. Dermal thickness, elasticity and blood vessel density can vary with factors such as age, sex and anatomic location [33, 34]. Sporozoites must migrate through this environment to reach blood vessels and a previous study found that sporozoite movement differs in both speed and type of trajectory between the ear and the tail of a mouse [35]. To test whether the location of mosquito probing could impact the likelihood of infection, we performed single mosquito feeds on two anatomical locations in addition to the ear, namely the tail and abdomen. As shown in Fig 4A, there is no statistically significant difference in the probability of infection when mosquitoes feed on different anatomical locations (p = 0.655, Fisher's exact test). Importantly, each group had similar numbers of mosquitoes with salivary gland loads over 10,000 (ear 37%, abdomen 25%, tail 31%). However, we observed that mosquitoes placed on the tail found blood more quickly compared to those on the abdomen or ear. Consistent with this observation, probe times were 45% and 50% shorter for mosquitoes placed on the tail, compared to mosquitoes placed on the abdomen and ear, respectively (Fig 4B, p<0.001). Nonetheless, their overall success rate in finding blood was not statistically significantly increased, with 74% of mosquitoes placed on the tail taking a

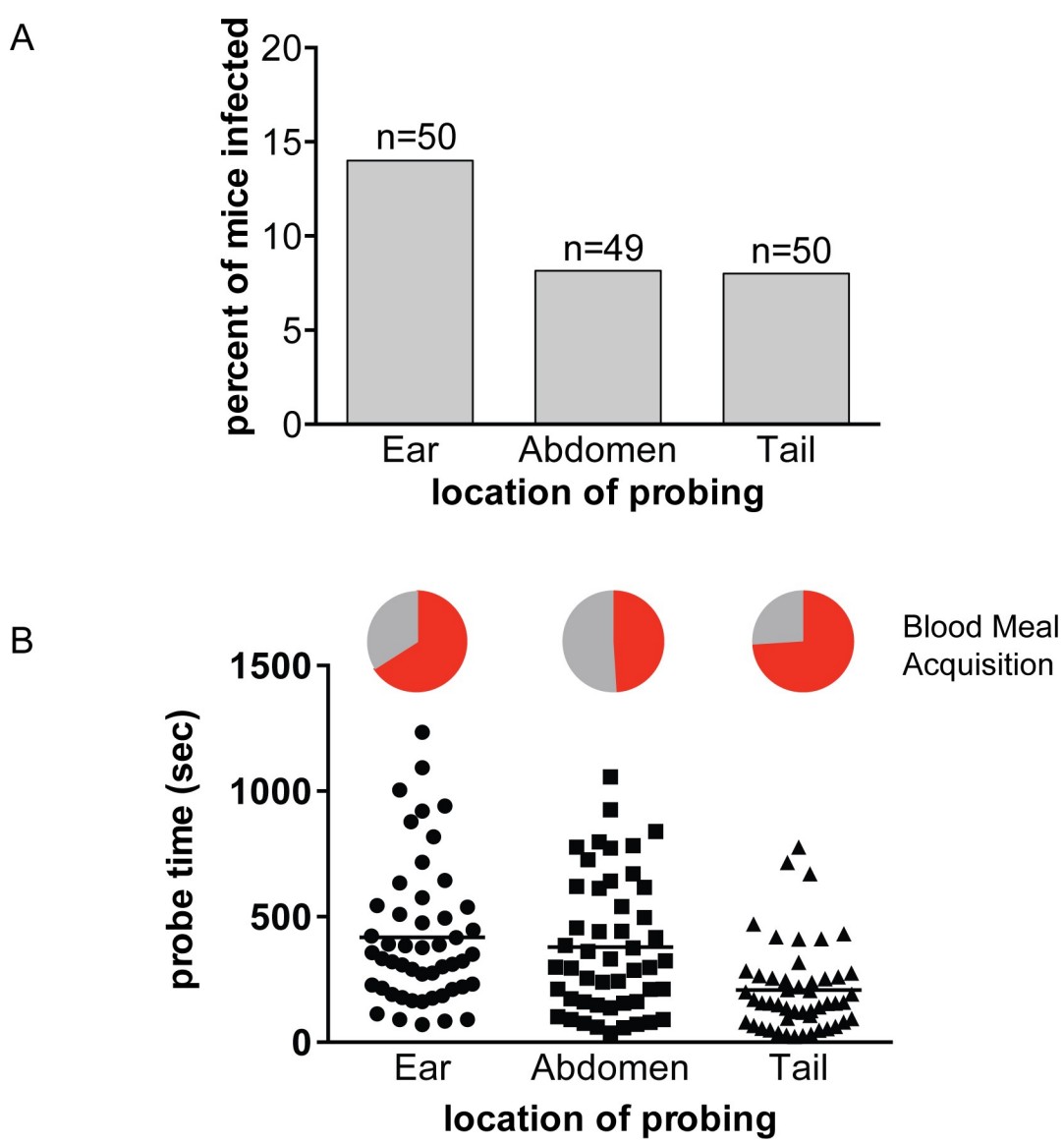

**Fig 4. The anatomical location of mosquito probing does not impact the likelihood of malaria infection.** Single-mosquito feeds were performed on the ear, abdomen or tail of mice and probe time and blood meal acquisition were recorded. Following this, salivary gland sporozoite loads were measured and mice were followed for blood stage malaria infection. **(A) Percent of mice positive for malaria infection when probed-upon at the indicated location.** n = total number of mouse-mosquito pairs in each group, pooled from 4 independent experiments with 10 to 15 mouse-mosquito pairs per group per experiment. The risk of infection is not statistically different between the three locations (p = 0.655). **(B) Probe time and blood meal acquisition of mosquitoes placed on different anatomical locations.** During the location feeds, mosquitoes were allowed to probe until they took a blood meal or became disinterested and flew away. Times to either of these endpoints were measured and were lower for mosquitoes placed on the tail compared to the abdomen and ear, by 45 to 50 percent respectively (p<0.001). Pie Charts: The percent of mosquitoes feeding on the ear, abdomen and tail that succeeded in obtaining a blood meal is shown in red and was 66%, 51%, and 74%, respectively. This difference did not reach statistical significance (p = 0.06, Fisher's exact test).

blood meal compared to 51% and 66% of those on the abdomen and ear, respectively (Fig 4B, pie charts; p = 0.06, Fisher's exact test). A previous study in guinea pigs measuring probe time and success in finding blood by *Aedes aegypti* mosquitoes found that mosquitoes probing on the back took significantly longer to find blood than those probing on the ear [25].

Interestingly, together with our study, these findings suggest that mosquitoes take longer to find blood on anatomical locations with more hair. Importantly, the *Aedes aegypti* study did not use infected mosquitoes and thus could not determine whether location of probing/feeding impacted the likelihood of initiating an infection. In our study using the ear, abdomen and tail, we find that the ability of mosquitoes to initiate a malaria infection does not correlate with where they feed.

## Likelihood of bloodmeal acquisition as it relates to salivary gland load and probe time

The goals of the *Plasmodium* parasite and its mosquito host are aligned in that the blood feeding behavior of the female Anopheline enables both mosquito reproduction and parasite transmission. However, the mosquito benefits when it obtains blood quickly, while the parasite benefits when probe time is longer or when the mosquito does not obtain a blood meal, which would necessitate probing on more than one host. Our mosquito feed data enabled us to analyze the association between these variables, using the 221 mosquito feeds in which probe time was not controlled, but was observed and recorded. First we determined whether there was an association between salivary gland sporozoite load and duration of probe time, and found no significant relationship between these two variables (Fig 5A; Spearman rank $\rho$ = -0.03, p = 0.66), in agreement with a previous study in which the intensity of mosquito infection did not impact probe time [36]. Next, we determined whether sporozoite salivary gland load was associated with the likelihood of blood meal acquisition and found that mosquitoes with high salivary gland sporozoite loads were less likely to obtain a blood meal (Fig 5B; Spearman rank $\rho$ = −0.2, p = 0.003). Logistic regression of blood meal acquisition also demonstrated a negative correlation with salivary gland sporozoite load, with the odds of obtaining a blood meal decreasing by 13% for every increase of 10,000 salivary gland sporozoites (95% CI: 5% to 21%, p-value = 0.002; Fig 5C).

## Discussion

The Ross-MacDonald model of malaria transmission serves as a basis for intervention strategies and new thinking about the quantitative aspects of malaria transmission [8]. Embedded within this model is the entomological inoculation rate (EIR), which quantifies human exposure to infected mosquitoes using mosquito density, biting rate and the rate of sporozoite positivity (Box 1). Early on, malariologists such as Draper, MacDonald, Davey and Gordon noted a "great discrepancy" between human exposure to infected mosquitoes (EIR) and the incidence of malaria infection [7, 9, 10, 37]. While the EIR estimates the *risk* encountered by individuals at various levels of transmission intensity, quantifying the proportion of infected bites that result in a blood stage infection allows for measurement of the 'force of human infection'. Thus, a proportional factor "b" was added to EIR (Box 1) where "b" is the fraction of infected human-feeding mosquitoes that initiate a malaria infection. Though "b" cannot be directly determined in humans, several studies have compared the incidence rate of malaria in infants to the biting rate of infected mosquitoes to arrive at estimates of "b" that range from 1 to 10% [7, 10, 11]. Here, using a rodent malaria model we provide the first experimental determination of "b", showing that 17% of mice have a patent blood stage malaria infection after a single infected mosquito bite. This infection rate of 17% is somewhat higher than what has been estimated in humans, and this is likely because our study was performed in naïve mice whereas infants in malaria-endemic areas have some passively-acquired immunity from maternal antibodies. Additionally, the proportion of highly infected mosquitoes in our study was higher than it is in the field [38–41] and as we discuss here, "b" is a function of mosquito salivary

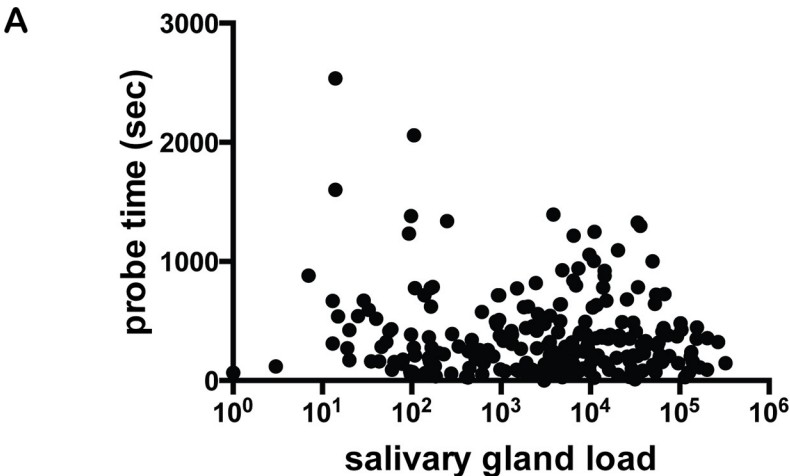

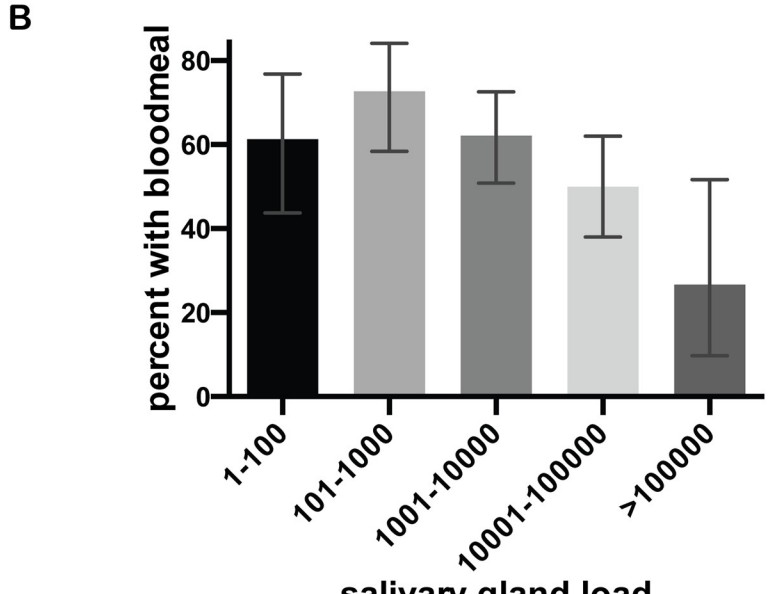

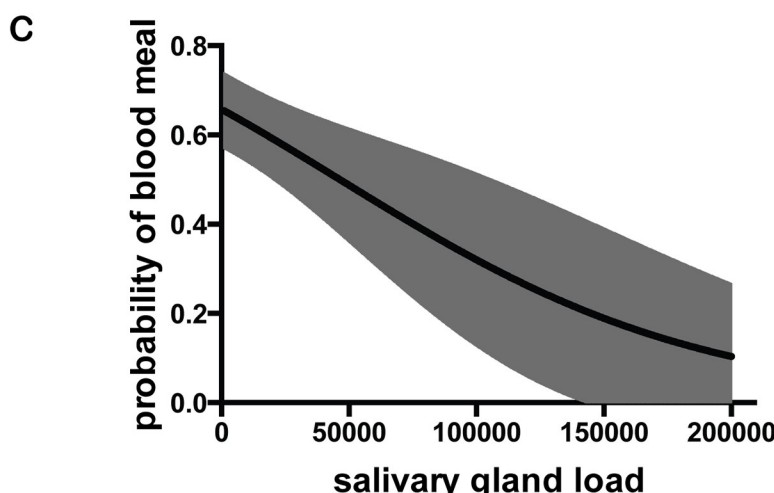

**Fig 5. The impact of salivary gland sporozoite load on probe time and blood meal acquisition.** Probe time and acquisition of blood meal were recorded for each mosquito feed, and subsequently salivary gland sporozoite loads were measured. (n = 221 mouse-mosquito pairs). **(A) Relationship of salivary gland sporozoite load and mosquito probe time.** For each mosquito, these parameters were plotted and correlation analysis revealed no significant association (Spearman rank $\rho$ = -0.03, p = 0.66). **(B) Binned salivary gland sporozoite loads suggest a relationship to blood meal acquisition.** Mosquitoes are binned using the traditional method of grading salivary gland infections on a log-scale and for each bin the percent of mosquitoes obtaining a blood meal is shown. Error bars show 95% confidence intervals calculated using Jefferey's intervals for binomial distribution [66]. **(C) Logistic regression analysis of blood meal acquisition as it relates to salivary gland sporozoite load.** Predicted probability of blood meal acquisition as it relates to salivary gland load from logistic regression is plotted against mosquito sporozoite load with 95% confidence intervals shown in gray shading. This analysis indicates that for every increase of 10,000 salivary gland sporozoites, the odds of obtaining a blood meal goes down by 13% (95% CI: from 5% to 21%, p-value = 0.002).

gland load and would be expected to vary depending on the intensity of mosquito infection. Given these caveats, our estimation of "b" is in the range of field estimates and demonstrates that the majority of infective mosquito bites do not result in a blood stage malaria infection.

Historically, the discrepancy between the biting rate of infected mosquitoes and the infection rate in humans was not well understood and was attributed to unknown immune factors that enabled infants to suppress the majority of sporozoite inoculations before parasitemia occurs [7, 10]. In our study the use of naïve mice eliminates the confounding influence of immunity and suggests that there are other explanation(s) for this phenomenon. Indeed work on pre-erythrocytic stages of *Plasmodium* over the past 20 years suggests that sporozoites face significant sequential bottlenecks between their development in the mosquito and establishment of infection in the mammalian host. For example, though mosquito salivary glands harbor thousands of sporozoites, the inoculum is generally low, in the range of 10 to 100 sporozoites, [22, 26, 27, 29, 30, 42, 43, 44]. Following their inoculation, only ~20% of sporozoites successfully exit the inoculation site [23, 24]. Furthermore, once in the liver some sporozoites get destroyed by Kupffer cells [45] and sporozoites that successfully enter hepatocytes face an innate immune response that may limit their success [46, 47]. Our data demonstrating that only a small fraction of infectious mosquito bites result in malaria infection is likely explained by the cumulative effect of successive bottlenecks that accompany this transmission phase.

Here we provide a direct demonstration that the intensity of infection in the mosquito has an impact on the probability of infection. The most likely explanation for this observation is that high sporozoite salivary gland loads translate into higher sporozoite inocula. Though our data do not enable us to reach this conclusion, since we could not simultaneously measure mosquito salivary gland loads and sporozoite inocula, previous studies found that highly infected mosquitoes were more likely to inoculate higher numbers of sporozoites [22, 43, 48]. It is also possible that at higher salivary gland loads, sporozoites become more infectious for the mammalian host, perhaps due to changes that occur when high numbers of sporozoites are tightly packed in the salivary glands [48]. Future studies looking at sporozoite inoculum size and sporozoite infectivity as it relates to mosquito salivary gland load should help clarify this question.

Our findings demonstrate that the relationship between mosquito salivary gland sporozoite load and infection probability is not linear. Indeed, logistic regression analysis, threshold modeling and binning all suggest that infection probability significantly increases at salivary gland loads between 10,000 to 20,000 sporozoites. Furthermore, we found that any model incorporating a rapid change in infection likelihood between 10,000 to 20,000 salivary gland sporozoites fits our data significantly better than the continuous models we tested. Thus the overall shape described by our data is a steep incline between salivary gland loads of 10,000 to 20,000 sporozoites, with relatively flat lines on either side of this incline. Though we do not yet understand the biological basis for the rapid increase, it may be related to previous

observations that sporozoites can be delivered as quanta [29, 32], which may occur at high sporozoite densities and increase infection likelihood. At salivary gland loads greater than the threshold, infection probability increases only marginally and appears to plateau at ~40%. While we do not understand why highly infected mosquitoes are not capable of initiating infection 100% of the time, several investigators have observed that highly infected mosquitoes inconsistently inoculate large numbers of sporozoites [22, 27, 29]. Indeed, in a previous study we found that the subset of highly infected mosquitoes inoculated >100 sporozoites about 40% of the time [22], a finding that is consistent with the frequency with which these mosquitoes initiate infection, suggesting that a threshold number of ~100 sporozoites may be required to reliably initiate infection. At low salivary gland sporozoite loads, our data suggest that the threshold inoculum is rarely met and infection may be a chance event. Given the low likelihood of infection at sporozoite loads <10,000 and the rapid rise in infection likelihood at sporozoite loads >10,000, we hypothesize that measuring the number of mosquitoes with gland loads >10,000 sporozoites in a given geographic area, could enable a more accurate determination of the force of infection.

Given the strong association between salivary gland sporozoite load and the likelihood of malaria infection in the rodent model, it is important to have some understanding of how these data relate to a field setting. Quantification of salivary gland sporozoite loads in wild-caught mosquitoes by investigators working in different regions of sub-Saharan Africa found that the percentage of infected mosquitoes with sporozoites loads over 10,000 ranges from 5–25%, with most studies putting this number at ~10% [27, 38–40, 49, 50]. Thus, our data suggest that bites from 75–95% of infected mosquitoes have a low probability of initiating a malaria infection after a single encounter with a human host. Though our experimental setup did not address the possibility of encounters with multiple hosts, we did find that mosquitoes with high sporozoite loads were less likely to find blood. In the field, the average number of hosts that are probed upon by infected mosquitoes prior to successful acquisition of blood is not known. Our data suggest that highly infected mosquitoes would be more likely to feed on multiple hosts and together with our finding that highly infected mosquitoes are more likely to initiate a malaria infection, we hypothesize that the small group of highly infected mosquitoes may contribute disproportionately to malaria transmission.

A recent study used data from human vaccine trials and rodent malaria transmission blocking studies to determine whether there was a relationship between mosquito salivary load and infection [51]. Both mice and humans in these studies were subjected to multiple mosquito bites, between 5–12 in the human studies and 1–10 in the rodent studies. Thus it is not possible to directly determine the probability that a single infected mosquito bite will initiate malaria infection. Nonetheless mathematical modeling of these data suggested that mosquitoes with ≤10 sporozoites initiate infection 32% and mosquitoes with >1000 sporozoites initiate infection 78% of the time: These numbers are 3 to 6 times higher than the infection probabilities we experimentally determined and likewise different from the infection probabilities estimated from field studies [7, 10, 11]. Several factors are likely responsible for this overestimation. Most importantly, the Churcher analysis [51] is based on data generated from vaccine trials, in which test subjects are challenged with a specific number of infected mosquito bites where "bite" is defined by a successful blood meal. Though subjects are exposed to both mosquitoes that find blood and mosquitoes that probe but do not find blood, only those that blood feed are included in the data analysis. Here we show that mosquitoes that probe but do not take a blood meal initiate malaria infection as efficiently as those that obtain a blood meal. Thus, the inclusion of mosquitoes that only probed in the experimental set-up combined with their exclusion in the subsequent analysis would lead to an overestimation of infection probability. Furthermore, the limited range of sporozoite salivary gland loads in their analysis, with all

mosquitoes harboring >1,000 salivary gland sporozoites being placed in the same bin, limits the utility of their data. We found a sharp increase in infection likelihood at salivary gland loads between 10,000–20,000 sporozoites and this would not be captured if all gland loads >1,000 sporozoites are grouped together. Thus, with their parameters, the shape of the infection curve as it relates to salivary gland sporozoite load cannot be accurately described. Importantly, since their cutoff of 1000 sporozoites corresponds roughly to the presence of one oocyst, their data can only tell us that mosquitoes with more than 1 oocyst have a greater likelihood of initiating a malaria infection than mosquitoes with a single oocyst. In contrast, our data suggest a more complex relationship between the level of mosquito infection and the likelihood of malaria infection and this is highly relevant in both field and research settings, where interventions aimed at reducing the number of oocysts are being tested. Whilst the Churcher study suggests that near complete elimination of oocysts is required, our data suggest that this may not be necessary to significantly impact transmission.

Previous studies in *Aedes* mosquitoes infected with *Plasmodium gallinaceum* sporozoites found that infected mosquitoes probed longer than uninfected mosquitoes and that this was due to parasite-mediated inhibition of mosquito saliva apyrase, a protein with anti-coagulant properties [31, 52, 53]. Though we did not compare probe times between uninfected and infected mosquitoes, in our study the probe time of infected mosquitoes was not correlated to salivary gland sporozoite load. Thus, while infection per se may impact probe time, the magnitude of salivary gland infection does not. We did, however, find an association between probe time and the likelihood of malaria infection. In the controlled probe time experiments, there was a statistically significant trend of increasing likelihood of infection with increased probe time, with the most consistently observed significant pairwise comparison being between mosquitoes that probed for 10 sec versus 5 min. A noteworthy observation from these experiments is that exceptionally short probe times are capable of initiating blood stage infection, further supporting the finding that blood meal acquisition is unnecessary for successful infection.

Lastly, our finding that blood meal acquisition has no impact on the likelihood of malaria infection has implications for the human malaria challenge model used in Phase I and II vaccine trials. Vaccinated and control volunteers are typically challenged by the bites of 5 infected mosquitoes, and acquisition of a blood meal is used as the endpoint for a successful encounter [54–56]. Thus, mosquitoes that probe but do not imbibe blood are replaced until a total of five have imbibed blood, resulting in a wide range of exposure to infected mosquitoes among the volunteers. Indeed the majority of volunteers must be exposed to additional infected mosquitoes so that a total of 5 have imbibed blood. Since an infected mosquito is equally likely to initiate a malaria infection whether or not it has imbibed blood, this practice means that volunteers in vaccine trials are not challenged with equivalent numbers of infected mosquitoes, thus creating discrepancies in the challenge dose among volunteers [54–56]. The blood meal endpoint is simple and indisputable, whereas mosquito probing requires direct observation of mosquitoes on each volunteer. Though observation of probing is more laborious and would require additional training of staff, our results suggest that changes in vaccine trial protocol should be considered.

This first estimate of the probability of infection after a single infectious bite in a naïve host highlights many aspects of malaria transmission that are yet unexplored and provides a long-overdue contribution to malaria transmission dynamics. Our data have implications for malaria control efforts and emphasize the need for quantitative measures of the intensity of the sporozoite rate in field populations. Indeed, it is possible that the small group of highly infected mosquitoes could play an outsized role in malaria transmission, making transmission blocking interventions that decrease the intensity of infection in the mosquito potentially as important as those that decrease prevalence of infection.

## Methods

### Ethics Statement

This work was done in accordance with recommendations in the Guide for the Care and Use of Laboratory Animals and the National Institutes of Health. All animal work was approved by the Johns Hopkins University Animal Care and Use Committee (protocols #M011H467 and #M014H363), which is fully accredited by the Association for the Assessment and Accreditation of Laboratory Animal Care.

### Mosquitoes and Mice

*Anopheles stephensi* mosquitoes were reared in the insectary at the Johns Hopkins Malaria Research Institute and infected with *Plasmodium yoelii* as described previously [57]. To ensure a range of salivary gland loads, in some experiments mosquitoes were fed on mice injected with blood diluted to 0.5% parasitemia instead of 1%. All experiments were performed between days 14 and 16 after the infected blood meal. Female Swiss Webster mice were purchased from Taconic Farms (Derwood, MD) and housed in the animal facility at the Johns Hopkins Bloomberg School of Public Health. The age of the mice ranged from 4 to 9 weeks old. Within each experiment, the same age and batch of mice was used.

### Single mosquito feeds

Each cage of *P. yoelii* infected mosquitoes was checked 10 days after an infected blood meal for prevalence of infection. Midguts were removed from ~20 mosquitoes, and oocyst numbers were estimated by phase-contrast microscopy. Only cages where ≥70% of the mosquitoes were infected were used. Single mosquito feeds were performed as previously described [22]: On day 14 to 16 after the infected blood meal, mosquitoes were anesthetized at 4˚C and sorted into individual plastic tubes 1 cm in diameter which contained mesh netting at one end. After securing the open end with Parafilm, the mosquitoes were returned to the incubator and were deprived of sugar water overnight. On the day of the experiment, mice were lightly anesthetized by intraperitoneal injection of ketamine (35–100 μg/g) and xylazine (6–15 μg/g) and placed on a slide warmer maintained at 37˚C to prevent a drop in body temperature due to anesthesia. A feeder containing a starved mosquito was placed on the ear of the mouse in such a way that the mosquito could only access the ear or, when indicated, the abdomen or tail. The duration of probing was recorded as the cumulative time that the mosquito was on the mouse with the proboscis moving in and out of the skin, ending with the visible start of the blood meal or when the mosquito lost interest in feeding, defined by spending 3 or more minutes flying about without re-alighting on the mouse. Mosquitoes were given access to mice for 30 minutes, though in 2 cases, mosquitoes that were still probing at 30 minutes were allowed to continue probing until they desisted. Mosquitoes showing no initial interest in feeding, i.e. flying around the tube and not landing on the mouse for >5 minutes, were removed and the mouse was exposed to a replacement mosquito. The acquisition of a blood meal was determined by observing the abdomen of the mosquito for engorgement and red coloration, and confirmed by noting the presence of blood in the esophagus during later dissection using a stereo microscope (10X magnification). After completion of the feed, the mosquito was placed on ice and salivary glands were removed for sporozoite quantification. Mice probed upon by uninfected mosquitoes were removed from the study and were not reused, while mice probed-upon by infected mosquitoes were observed for the presence of blood stage infection by Giemsa-stained blood smears on days 5, 10 and 15 post-feed. All mice that became positive for

blood stage infection were positive on day 5 and the smears on days 10 and 15 confirmed the presence of parasites.

Controlled probe time experiments were conducted by allowing a mosquito to probe on the ear of a mouse for exactly 10 seconds, 1 minute, or 5 minutes. Timing was begun as soon as the proboscis entered the skin, and the feeder was removed after the designated time had elapsed. In instances where the mosquito halted probing prior to the end of the designated probe time (one minute and five minute probes only), the mosquito remained under observation until probing resumed or, if the complete probing duration was not met, both mosquito and mouse were removed from the analysis. In total, three of the mice in the 5 minute probe time group were removed because the mosquitoes did not complete the full five minutes. Occasionally, mosquitoes halted probing and began acquiring a blood meal before the designated probe time had elapsed (five minute probe condition only); in these instances the feeder was gently lifted to disengage the proboscis and the replaced on the ear, forcing the mosquito to resume probing. The timer was paused from the start of blood meal acquisition to restart of probing to ensure that the cumulative probe time was of the required duration.

For studies investigating the effect of location on transmission efficiency, the feeder was placed as follows: For bites to the tail the feeder was placed approximately half way down the length of the tail and for bites to the abdomen the feeder was centered on the ventral surface of the abdomen.

### Quantification of sporozoite salivary gland load

Following the experimental procedures described in the previous section, mosquitoes were dissected and the salivary glands collected. Genomic DNA (gDNA) was isolated from salivary glands using Qiagen DNeasy Kit (#69504, Qiagen Inc) according to manufacturer's recommendations, with two minor changes: The incubation time with proteinase K was increased to 30 minutes instead of the recommended 10, to allow for breakdown of excess mosquito material, and the DNA elution volume was reduced from 200 µL to two elutions of 20 µL in order to increase the DNA concentration and maximize yield. Genomic DNA samples were stored at -80˚C until quantification by quantitative PCR (qPCR).

A standard curve was made by isolating gDNA from known numbers of sporozoites, counted using a haemocytometer. A serial dilution of the sporozoites was performed to yield a log-fold dilution of sporozoites from 50 to $5 \times 10^4$ or when necessary 50 to $10^6$, to cover the range of salivary gland loads in experimental mosquitoes (see S1 Fig). Aliquots of known numbers of sporozoites were frozen at -80˚C until they were processed with the experimental salivary glands, as described above. Standard curves with known numbers of sporozoites were verified with a plasmid standard curve, using serial dilutions of a plasmid containing a fragment of the 18S rRNA C-type gene. Plasmid copy number was determined by calculating the plasmid size and mass of a single plasmid molecule, and multiplying this by the copy number of interest to calculate the concentration of plasmid DNA needed to achieve the desired copy number. We then prepared this solution (usually at $10^{10}$ copies in 4 microlitres) froze aliquots and used these to perform serial dilutions of the plasmid.

qPCR was performed on the gDNA isolated from the experimental mosquito salivary glands with a sporozoite standard curve for each experiment. Primers and cycling profile used have been previously described [3]. Briefly, primers specific for *P. yoelii* 18S ribosomal C-type RNA (forward primer, 5'-GGGGATTGGTTTTGACGTTTTTGCG-3' and reverse primer, 5'AAGCATTAAATAAAGCG AATACATCCTTAT-3') and 4 µL gDNA in a total volume of 25 µL/well was used and qPCR was performed with the StepOnePlus™ system (Applied Biosystems, Carlsbad, CA) using SYBR® Green PCR Mastermix (Life Technologies, Grand Island,

NY). The cycling profile was 95˚C for 15 min followed by 40 cycles of: 95˚C, 20s; 60˚C, 60s. After amplification, the melting temperature was determined using a dissociation curve to ensure that a single, specific product was formed. The profile for the melt curve was: 95˚C for 15s, 60˚C for 60s, and incremental increases of 3˚C up to 95˚C.

## Mathematical modeling

To understand the relationship between sporozoite salivary gland load and the probability of malaria infection we used several alternative mathematical models [58–61]. The simplest, "single-hit" model assumes that sporozoites act independently and that even one parasite in the salivary glands is capable of causing blood stage infection. In this model the relationship between the probability of infection and the number of sporozoites $S$ is:

$$p(S) = 1 - e^{-\lambda S} \tag{1}$$

where $\lambda$ is a parameter relating infection probability and sporozoite loads. In many cases, however, the single hit model is inconsistent with experimental data on infection (e.g., see [60]), for example when change in the probability of infection with the number of sporozoites changes more slowly than a simple exponential function. The "powerlaw" model therefore considers the change in the probability of infection as it depends on the "cooperativity" or "competition" between parasites indicated by the parameter $n$:

$$p(S) = 1 - e^{-\lambda S^n} \tag{2}$$

where n<1 indicates potential competition between parasites. We also considered a model in which the probability of infection is described by a step-like function ("threshold" model):

$$p(S) = \begin{cases} p_{min}, & \text{if } S < S^*, \\ p_{max}, & \text{otherwise,} \end{cases} \tag{3}$$

where $S^*$ is the threshold sporozoite number, $p_{min}$ and $p_{max}$ are the infection probabilities for sporozoite salivary gland load when it is below and above the threshold $S^*$, respectively.

We also tested two alternative threshold models. The first, a slope-threshold model extends *Eq* 3 by including a linear increase in the infection probability with sporozoite load at low sporozoite values $\alpha$:

$$p(S) = \begin{cases} p_{min} + \alpha S, & \text{if } S < S^*, \\ p_{max}, & \text{otherwise,} \end{cases} \tag{4}$$

Note that this model can be discontinuous at $S = S^*$. Second, we consider a model in which infection probability changes as double-logistic function:

$$p(S) = \begin{cases} \dfrac{p_m}{p_m + e^{-\lambda_1 S}}, & \text{if } S < S^*, \\ \dfrac{p_m}{p_m + e^{-\lambda_1 S^* - \lambda_2 (S - S^*)}}, & \text{otherwise,} \end{cases} \tag{5}$$

where $p_m$, $\lambda_1 \lambda_2$, and $S^*$ are model parameters. Note that when $\lambda_2 = 0$ and $S^*$ is large, this model reduces to the standard logistic model.

To fit these mathematical models [*Eq*s (1)–(5)] to experimental data we use the likelihood approach where the likelihood of the model given the data is:

$$L \sim \prod_{i=1}^{N} p(S_i)^{D_i} (1 - p(S_i))^{1 - D_i} \tag{6}$$

where N is the number of mosquitoes, $p(S_i)$ is given in *Eqs* (1)–(5), $S_i$ is the sporozoite number in salivary glands of an $i^{th}$ mosquito, and $D_i = (0,1)$ is the probability that bite by an $i^{th}$ mosquito leads to blood stage infection. Parameters estimated by fitting models to infection probability data are as follows: single-hit ($\lambda = 5.8 \times 10^{-6}$), powerlaw ($\lambda = 3.8 \times 10^{-3}, n = 0.41$), threshold ($p_{min} = 0.066$, $p_{max} = 0.35$, $S^* = 20166$), slope-threshold ($p_{min} = 0.0404$, $p_{max} = 0.35$, $S^* = 20242$, $\alpha = 6.06 \times 10^{-6}$), and double logistic ($p_m = 0.041$, $\lambda_1 = 1.10 \times 10^{-4}$, $\lambda_2 = 3.26 \times 10^{-6}$, $S^* = 20737$). Given that three alternative models (threshold, slope-threshold, and double logistic) provided the best fits of the data, we also calculated 95% confidence intervals for best fit parameters by resampling the data with replacement 1,000 times: threshold model (*Eq* (3)): $p_{min} = (0.0375, 0.0985)$, $p_{max} = (0.280, 0.430)$, $S^* = (19368, 21408)$; slope-threshold model (*Eq* (4)): $p_{min} = (0.012, 0.077)$, $p_{max} = (0.280, 0.430)$, $\alpha = (0.0, 1.34) \times 10^{-5}$, $S^* = (16967, 27404)$; double-logistic model (*Eq* (5)): $p_m = (0.016, 0.072)$, $\lambda_1 = (0.60, 1.83) \times 10^{-4}$, $\lambda_2 = (0.0, 7.82) \times 10^{-6}$, $S^* = (11099, 33801)$. To compare fit quality of different models we calculated corrected Akaike Information Criterion (AIC) and Akaike weights [62]. Akaike weight approximately represents the "likelihood" of a given model in the list of tested models. To compare nested models we used the likelihood ratio test [63] and to compare the quality of model fits to experimental data we used the Hosmer-Lemeshow test, binning the data into 7 or 8 bins.

## Statistical analyses

Logistic regression models with probability of malaria infection as the dependent variable were used to assess the associations with blood meal status, location of the bite, probe time, and sporozoite load. All models included robust variance estimates to adjust for potential within-experiment correlations of malaria infections [64]. Categorical variables, such as location or probe time were represented as indicator variables in the model. P-values from Wald tests for the corresponding beta coefficients are reported. Visual displays were used to assess the relationship between continuous predictors, such as sporozoite load and malaria infection. Log-transformations and linear splines were used to represent deviations from linearity in relationship between these continuous predictors and the dependent variable. Correlations between measurements in 5A were performed using Spearman rank correlation coefficient. All tests were two-sided and were carried out at 5% level of statistical significance. Stata 14 was used in these analyses [65].

For the probe time experiments, the relationship between probe time and malaria infection was analyzed using the Fisher's exact test. Following this, pairwise comparisons were performed using logistic regression analysis where malaria infection was the dependent variable and probe time was the covariat, represented by 2 indicator variables (1 min and 5 min probe time groups) with the 10 sec probe time group being the reference. The p-value for the slope of each group represents the comparison of that group to the reference. Comparison of the 2 slopes gives the p-value for the 1 min versus the 5 min groups. The model included a robust variance estimate to account for potential correlation between malaria risk within the same experiment.

To assess the relationship between probability of malaria infection ($P$) and sporozoite load ($S$), logistic regression model was fit to the data. The results of the exploratory analyses indicated a non-linear relationship between sporozoite load and log-odds of malaria infection. This non-linear relationship is represented with linear spline of sporozoite load with a knot at 20,000 copies (i.e. piece-wise linear slope model). The model is formulated as follows:

$$P(S) = \frac{\exp(\beta_0 + S_1\beta_1 + S_2\beta_2)}{1 + \exp(\beta_0 + S_1\beta_1 + S_2\beta_2)}, \tag{7}$$

where $S_1 < 20,000$ copies and $S_2 \geq 20,000$ copies, $\beta_0$ is the log-odds of malaria infection with no sporozoite load (i.e. the intercept), $\beta_1$ is the change in log-odds of malaria infection for every 500 copies increase in sporozoite load up to 20,000 copies (i.e. slope for $S_1 < 20,000$ copies), and $\beta_2$ is the change in log-odds of malaria infection for every 500 copies increase in sporozoite load between 20,000 and 647,714 copies (i.e. slope for $S_2 \geq 20,000$ copies). After obtaining maximum likelihood estimates [65, 66] for $\beta_0$, $\beta_1$ and $\beta_2$, the regression equation is as follows:

$$\hat{P}(S) = \frac{\exp(-3.2 + 0.057S_1 + 0.002S_2)}{1 + \exp(-3.2 + 0.057S_1 + 0.002S_2)}, \tag{8}$$

where $S_1$ and $S_2$ are the number of sporozoites in units of 500.

Logistic regression was also used to model the relationship between the probability of blood meal ($\pi$) and the number of sporozoites (S). The model was formulated as follows:

$$\pi = \frac{\exp(\beta_0 + S\beta)}{1 + \exp(\beta_0 + S\beta)}, \tag{9}$$

where S is the number of sporozoite copies in units of 500, $\beta_0$ is the log-odds of blood meal with no sporozoite load (i.e. the intercept), $\beta_1$ is the change in log-odds of blood meal for every 500 copies increase in sporozoite load (i.e. slope for sporozoite load). After obtaining maximum likelihood [66, 67] estimates for $\beta_0$ and $\beta_1$, the regression equation is as follows:

$$\hat{\pi}(S) = \frac{\exp(0.653 - 0.007S)}{1 + \exp(0.653 - 0.007S)}, \tag{10}$$

## Supporting information

**S1 Fig. Distribution of salivary gland sporozoite loads in the 412 mosquitoes used in this study (A) and qPCR standard curve used to determine salivary gland loads (B).** (A) Salivary gland loads are binned as indicated in the X-axis and shown is the number of mosquitoes that fall into each bin. The median salivary gland load is 8,865 sporozoites. 216 mosquitoes had <10,000 and 196 mosquitoes had >10,000 salivary gland sporozoites. (B) qPCR quantification of 10-fold dilutions of a known number of sporozoites using 18s rRNA plasmid copy numbers as a standard.
(PDF)

**S2 Fig. Probability of malaria infection as a function of salivary gland load using a logistic regression model with a linear spline at 20,000 salivary gland sporozoites.** The full dataset, extending from salivary gland loads of 1 to 647,714 sporozoites, is plotted with 95% confidence intervals for each point. The graph below displays the data from mosquitoes with salivary gland loads of 1 to 30,000 (boxed region of top graph) to better display the data in this range.
(PDF)

**S3 Fig. Infection probability as a function of sporozoite load using mathematical models incorporating a rapid change around a threshold.** We fit three different mathematical models that include a rapid increase in infection probability ("threshold", "slope-threshold", or "double-logistic") to the mouse-mosquito dataset (n = 408 mouse-mosquito pairs with salivary gland loads between 1 and 300,000 sporozoites), using the maximum likelihood method (see *Eq* 6 in Materials and Methods). The quality of the model fit to data was calculated using Akaike Information Criterion to generate Akaike weights (*w*). While the models incorporating

a gradual rise between salivary gland loads of 10,000 to 20,000 sporozoites provided a better fit based on $w$ values, the differences between the weights was small and the Hosmer-Lemeshow goodness of fit test indicated that all 3 models fit the data well ($p > 0.1$). The threshold, slope-threshold, and double-logistic models are described by *Eqs* 3–5, respectively, in the Materials and Methods. Parameters of the models providing the best fit are also given in the Materials and Methods.
(PDF)

**S4 Fig. Infection likelihood after bites of mosquitoes with salivary gland loads greater than or less than 10,000 sporozoites.** Single-mosquito feeds were performed, and subsequently salivary gland sporozoite loads were measured and mice were followed for blood stage malaria infection. Mosquitoes are binned as to whether they had greater than or less than 10,000 salivary gland sporozoites, and the probability that the mosquitoes in each group will initiate a malaria infection is plotted. Salivary gland sporozoite load is significantly associated with the likelihood of malaria infection (odds ratio 7.5, CI 3.6–15.8, p<0.001). n = total number of mouse-mosquito pairs in each bin, pooled from 20 independent experiments with 7 to 44 mouse-mosquito pairs per experiment, with total n = 412.
(PDF)

**S5 Fig. Predicted infection probability in a field setting based on the percentage of mosquitoes with salivary gland sporozoite loads of <10,000 sporozoites.** Shown is the decline in infection probability per mosquito bite as the fraction of mosquitoes with low (<10,000) sporozoite loads increases. In the full dataset of n = 412 mosquito feeds we found that probability of infection by mosquitoes with <10,000 sporozoite loads is $b_1 = 5.6\%$ while bite by a mosquito with $>10^4$ sporozoites results in infection with probability $b_2 = 30.6\%$. If the proportion of mosquitoes with low (<10,000) sporozoite numbers is $p$, then the probability of infection per bite by a mosquito in such a population is given by $b = p^* b_1 + (1-p)^* b_2$. Confidence intervals for these predictions were calculated using Jefferey's intervals for binomial proportions for estimated values $b_1$ and $b_2$.
(PDF)

**S1 Table. Raw data, including probe time, anatomical location, blood meal acquisition and blood stage infection from all mosquito-mouse pairs in this study.**
(XLSX)

## Acknowledgments

The authors would like to thank the Johns Hopkins Malaria Research Institute Insectary and Parasitology core facilities, especially Christopher Kizito for expert rearing of mosquitoes and Dr. Godfree Mlambo for production of the *P. yoelii* infected mosquitos. We would also like to thank members of the Sinnis Laboratory and Drs. Elizabeth Ogburn, David Sullivan and Fernando Pineda for their helpful and illuminating discussions.

## Author Contributions

**Conceptualization:** Maya Aleshnick, Photini Sinnis.

**Data curation:** Maya Aleshnick, Vitaly V. Ganusov, Gibran Nasir, Gayane Yenokyan, Photini Sinnis.

**Formal analysis:** Maya Aleshnick, Vitaly V. Ganusov, Gibran Nasir, Gayane Yenokyan, Photini Sinnis.

**Funding acquisition:** Photini Sinnis.

**Investigation:** Maya Aleshnick, Gibran Nasir.

**Methodology:** Maya Aleshnick, Gibran Nasir, Gayane Yenokyan, Photini Sinnis.

**Project administration:** Photini Sinnis.

**Supervision:** Photini Sinnis.

**Validation:** Maya Aleshnick, Vitaly V. Ganusov, Gibran Nasir, Gayane Yenokyan, Photini Sinnis.

**Writing – original draft:** Maya Aleshnick, Photini Sinnis.

**Writing – review & editing:** Maya Aleshnick, Vitaly V. Ganusov, Gibran Nasir, Gayane Yenokyan, Photini Sinnis.

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
