## [Decision Letter · Decision Letter 0]

16 Dec 2019

Dear Dr. Sinnis:

Thank you very much for submitting your manuscript "Experimental Determination of the Force of Malaria Infection Reveals a Non-Linear Relationship to Mosquito Sporozoite Loads" (PPATHOGENS-D-19-01962) for review by PLOS Pathogens. Your manuscript was fully evaluated at the editorial level and by independent peer reviewers. All three reviewers were excited by the paper, and appreciated the attention to an important topic, but identified aspects of the manuscript that should be improved.

We therefore ask you to modify the manuscript according to the review's recommendations before we can consider your manuscript for acceptance. Your revisions should address the specific points made by each reviewer. In particular, reviewer 2 raised several technical issues that need addressing and all reviewers made suggestions to improve clarity and impact.

(1) A letter containing a detailed list of your responses to the review comments and a description of the changes you have made in the manuscript. Please note while forming your response, if your article is accepted, you may have the opportunity to make the peer review history publicly available. The record will include editor decision letters (with reviews) and your responses to reviewer comments. If eligible, we will contact you to opt in or out.

(2) Two versions of the manuscript: one with either highlights or tracked changes denoting where the text has been changed; the other a clean version (uploaded as the manuscript file).

We hope to receive your revised manuscript within 60 days or less. If you anticipate any delay in its return, we ask that you let us know the expected resubmission date by replying to this email.

[LINK]

Sincerely,

Tim J.C. Anderson

Guest Editor

PLOS Pathogens

Kirk Deitsch

Section Editor

PLOS Pathogens

Kasturi Haldar

Editor-in-Chief

PLOS Pathogens

orcid.org/0000-0001-5065-158X

Grant McFadden

Editor-in-Chief

PLOS Pathogens

orcid.org/0000-0002-2556-3526

Reviewer's Responses to Questions

**Part I - Summary**

Reviewer #1: The manuscript by Aleshnick and colleagues is a very timely, important and robust paper. The data are unique and provide important insights in developmental bottlenecks for malaria parasites. Whilst a non-human malaria model is used, it is plausible that the findings also have implications for Plasmodium species that are relevant for humans. I have only minor comments.

Reviewer #2: This manuscript investigated the impact of Plasmodium sporozoite load, blood-meal acquisition, probe-time, and probe location, on malaria infection probability. The results showed that mosquitoes with higher salivary gland sporozoites are more likely to initiate malaria infection, which provides reference data of a new useful phenotype for field studies. They also proved that infection probability was not impacted by whether a blood meal had been acquired by the mosquito. Overall, this paper will be interesting for readership working on malaria control and vaccine development.

However, some methods used in this study are not very appropriate or not well described. For example, the qPCR primers indicated in this paper are able to amplify three locations of the Plasmodium yoelii yoelii 17XNL genome, which was not mentioned anywhere in the manuscript.

Reviewer #3: This is animportant manuscript which will have a strong impact on the field. In a tour-de-force experiments, the authors quantify the dynamics of Plasmodium transmission from mosquito to a mammalian host and reveal that only one fith of the infectious bites will result in malaria infections. The authors develop a model that explains non-linear relationships between the sporozoite loads in the mosquito salivary glands and transmission success, defining that malaria infections are transmitted by a small number of highly infected mosquitoes. These conclusions have important consequences for malaria epidemiology and for design of anti-sporozoite vaccine trials.

Although the number of sporozoites that are sufficient to initiate infection was not addressed in this manuscript, the authors provide strong evidence that only bites of mosquitoes that have more than 10,000 sporozoites will initiate the disease. Such quantitative approaches can be directly applied to the field studies instead of the currently used metrics of entomological innoculation rates that only consider the number of bites per person as a poxy for infectious bites.

Based on the novelty and significance of the presented data I strongly recommend this manuscipt for publication in PLoS Pathogens.

**Part II – Major Issues: Key Experiments Required for Acceptance**

Reviewer #1: None.

Reviewer #2: 1. qPCR

There will be three amplification products using the qPCR primers indicated in this paper, according to the Plasmodium yoelii yoelii 17XNL genome (https://plasmodb.org/common/downloads/release-46/Pyoeliiyoelii17XNL/). The templates are contig AABL01000525, AABL01001425 and AABL01002193. Normally, genes with multiple copies in the genome are not good candidate reference genes for qPCR analysis, due to the instability of such kind of genes. This should be mentioned in the manuscript.

The authors were using known numbers of sporozoites to generate standard curve, in which way the DNA recovery rate should be able obtained, especially the authors also used 18s rRNA plasmid as standard (Supplementary Figure 1). It’s better to include the DNA recovery rates for different numbers of sporozoites in the manuscript or in the Supplementary files. Method for measuring 18s rRNA plasmid copy numbers should also be included.

Also, a supplementary file/figure to show the primer amplification efficient and the standard curve would be useful for other people to re-use the data.

2. Controlled/uncontrolled probe time experiments

There were two mosquito feeding methods used in this study: controlled/uncontrolled probe time experiments. 1) In the controlled probe time experiments (page 29), some of the mosquitoes were forced to resume probing, which increased the probing by human manipulation. 2) Mosquitoes in the controlled probe time experiment groups also had more sporozoite load than in the controlled probe time experiment groups (Figure 3 legend). 3) The explore time of mosquito to mice were different in these two experiment groups: 30min for uncontrolled groups, while variety for the controlled groups. It seems there were multiple variances in these two experiments, should it be better to separate the analysis of these two experiments (Figure 2; page 8, parag 1; page 12, parag 2)?

Reviewer #3: I do not think that additional experiments are required.

**Part III – Minor Issues: Editorial and Data Presentation Modifications**

Reviewer #1: Abstract: ‘being 7.5 times more likely to initiate a malaria infection’ suggests a precision that is not supported by the data. I would suggest to rephrase this and make it more descriptive (e.g. ‘considerably more likely… compared to mosquitoes with lower infection burden’

Author summary: ‘In this study, using a rodent… results in malaria infection’ contains a duplication of messages. Minority infecting and majority non-infecting is clearly the same. I would suggest to simplify this.

I would strengthen the author summary by making not only a comment on the importance for interventions but also for understanding the epidemiology of malaria.

The introduction is very well written and nicely illustrates (very old) literature and current-day relevance of the question addressed.

The first section of the results explains why P. yoelii was chosen. Part of the argument is missing. I believe that the authors want to say that because yoelii has such a high likelihood of resulting in detectable infections, it provides a sensitive system to detect/quantify potentially infectious bites. It would be good to mention this specifically.

The direct observation of effective contact of mosquitoes is a considerable strength of the paper

Figure 1 and 5b should come with error bars

Page 8, ‘over 400’: give exact number.

Figure 2c. the step appears completely driven by one data point that lies well above the fitted line. It would be good to discuss this in the discussion. The data overall are very convincing but the strong claim on this step is a weaker part of the (overall very stron) paper. I am unconvinced the step would be observed again if the study would be repeated.

Fig3b is confusing. Too much attention is drawn to the first bar with a very low number of observations. It would be more appropriate to combine the two lower bins

Methods are great and very complete; the discussion is appropriate and well written. It is great to see the Ross McDonald model being criticized in a positive and data-supported manner.

Reviewer #2: Page2, Abstract: “sporozoites to patent blood-stage infection”, patient?

Page 6, Supplemental Table. An extra column will be much helpful to clarify to the experimental methods, other than by probe time values of (10, 60, and 300 seconds). Maybe also include Probe time (s) for mosquitoes without infection (Salivary Gland Load = 0).

Page 8 parag 2 & page 23 parag 2: what’s the range of sporozoite load in field samples?

Page 8 second last lane: add reference.

Page 9, page 12: The abbreviation for p value should be unified.

Page 12 & Figure 2: the infection probability seems to reach a plateau after 20,000 spz load, any possible explanations?

There were multiple logistic regressions mentioned in this manuscript, a summary of the parameters in these regressions should make the results much clearer.

Figure 3: The authors mentioned that spz load may explain the different patterns in panel A and B. An extra scatter/box diagram of the spz load in different bin would help to clarify. panel A, how many samples in the 5min group have been forced to resume probing? What’s the significant level between 10 sec and 1min group? panel B, there was no mouse infected by mosquito (more than 100 samples) with >5min probing, any reason? The spz loads between “>5-10 min” and “>1-5 min” were not significantly different while the infection rates were quite different (page 15), why?

Page 15: does “11,766 +/- 16,786” mean “range from 11,766 to 16,786”?

Figure 4: legend for pie chart.

Figure 5: it’s better to mention that it’s based on 30min of mosquito-mouse exposure experiment. Any differences in probe time and spz load, between mosquitoes that able to get a blood-meal and not able to get blood-meal in 30min?

Page 21 lane 4: what’s the proportion in this study and in the field?

Page 24: what’s the infection probabilities estimated from field studies?

Reviewer #3: To increase the clarity of the manuscript I would like the authors to restructure the information split between the Introduction and Discussion. Ross-MacDonald formula should be described in the Introduction to setup the stage for the current study. otherwise, the information appears in both places but is very imprecise in the Introduction which is cryptic for a wide audience of readers. Eliminating repetitions between the Results and Discussion will also make the stronger the Discussion part.

I was also confused by the discussion of the results presented in Figure 3A: "As shown in Figure 3A, mosquitoes that were allowed to probe longer were more likely to initiate a malaria infection (p=0.020, Fisher’s exact test) with pairwise comparisons showing a statistically significant difference in infection probability between the 10 sec and 5 min probe times (p=0.021) and a less robust but significant difference between the 1 min and 5 min probe times (p=0.025). The difference between the 1 min and 5 min groups was not significant by non-parametric tests such as the Mann-Whitney (p=0.15)."

Could the authors use the appropriate statistical tools? If the data is normally distributed, the use of parametric tests is justified. If the data is not normally distributed, only non-parametric tests should be used. The authors have to make clear whether the observed differences are statistically significant or not and show variability and p-values on the graph.

The Figure 3B is also confusing. If the authors identify problems in the dataset, it should not be presented in the main figure as such. Instead, the authors could only present the data for the groups that can be compared (for example, the group probing for 1-5 min and the group probing for 5-10 min).

PLOS authors have the option to publish the peer review history of their article (what does this mean?). If published, this will include your full peer review and any attached files.

Reviewer #1: No

Reviewer #2: Yes: Xue Li

Reviewer #3: No

---

## [Editor Report · Decision Letter 1]

4 Apr 2020

Dear Dr. Sinnis,

We are pleased to inform you that your manuscript 'Experimental Determination of the Force of Malaria Infection Reveals a Non-Linear Relationship to Mosquito Sporozoite Loads' has been provisionally accepted for publication in PLOS Pathogens.

Best regards,

Tim J.C. Anderson

Guest Editor

PLOS Pathogens

Kirk Deitsch

Section Editor

PLOS Pathogens

Kasturi Haldar

Editor-in-Chief

PLOS Pathogens

orcid.org/0000-0001-5065-158X

Michael Malim

Editor-in-Chief

PLOS Pathogens

orcid.org/0000-0002-7699-2064

I was delighted to see the reviewers comments so carefully and thoughtfully addressed. The paper is considerably improved and a valuable contribution.
---

## [Editor Report · Acceptance letter]

18 May 2020

Dear Dr. Sinnis,

We are delighted to inform you that your manuscript, "Experimental Determination of the Force of Malaria Infection Reveals a Non-Linear Relationship to Mosquito Sporozoite Loads," has been formally accepted for publication in PLOS Pathogens.

Best regards,

Kasturi Haldar

Editor-in-Chief

PLOS Pathogens

orcid.org/0000-0001-5065-158X

Michael Malim

Editor-in-Chief

PLOS Pathogens

orcid.org/0000-0002-7699-2064